# Higher Availability of Long-Chain Monounsaturated Fatty Acids in Preterm than in Full-Term Human Milk

**DOI:** 10.3390/life13051205

**Published:** 2023-05-17

**Authors:** Tamás Marosvölgyi, Timea Dergez, József L. Szentpéteri, Éva Szabó, Tamás Decsi

**Affiliations:** 1Institute of Bioanalysis, Medical School, University of Pécs, 7624 Pécs, Hungary; 2Department of Paediatrics, Medical School, University of Pécs, 7623 Pécs, Hungary; 3Institute of Transdisciplinary Discoveries, Medical School, University of Pécs, 7624 Pécs, Hungary; 4Department of Biochemistry and Medical Chemistry, Medical School, University of Pécs, 7624 Pécs, Hungary

**Keywords:** colostrum, erucic acid, gondoic acid, human milk, long-chain monounsaturated fatty acids (LCMUFAs), mature milk, nervonic acid, transient milk

## Abstract

While the role of n-3 and n-6 long-chain polyunsaturated fatty acids (LCPUFAs) in the maturation of the infantile nervous system is extensively studied and relatively well-characterized, data on the potential developmental importance of the n-9 long-chain monounsaturated fatty acid (LCMUFA), nervonic acid (NA, C24:1n-9) are scarce and ambiguous. Therefore, the aim of the present study was to reanalyze our available data on the contribution of NA and its LCMUFA precursors, gondoic acid (C20:1n-9) and erucic acid (EA, C22:1n-9) to the fatty acid composition of human milk (HM) during the first month of lactation in mothers of both preterm (PT) and full-term (FT) infants. HM samples were obtained daily during the first week of lactation, and then on the 14th, 21st, and 28th days. Values of the LCMUFAs, C20:1n-9, EA, and NA were significantly higher in colostrum than in transient and mature HM. Consequently, there were highly significant inverse associations between LCMUFA values and the duration of lactation. Moreover, C20:1n-9, EA, and NA values were monotonously, considerably, and at many timepoints significantly higher in PT than in FT HM samples. By the 28th day of lactation, summarized LCMUFA values in PT HM samples declined to the level measured in FT HM samples on the first day of lactation; however, EA and NA values were still significantly higher in PT than in FT HM on the 28th day. Significantly higher availability of LCMUFAs in PT than in FT HM underpins the potential biological role of this hitherto somewhat neglected group of fatty acids.

## 1. Introduction

The long-chain polyunsaturated fatty acids (LCPUFAs), arachidonic acid (AA, C20:4n-6), and docosahexaenoic acid (DHA, C22:6n-3) has been considered for a long time to play an important role in perinatal development, especially in the maturation process of the nervous system [1]. However, not only AA and DHA but also one of the monounsaturated fatty acids (MUFAs), nervonic acid (NA, C24:1n-9) is among the most important structural building blocks within the central nervous system [2]. AA and DHA contribute to structure formation in neuronal membranes, making up around 20% of the dry mass of the brain; nevertheless, in cerebellar white matter total lipids the content of NA more than doubles in breast-fed infants during the first 20 weeks of life [3].

While AA and DHA play a predominant role in the phosphoglycerides of early and near-term human placental membranes, more than half of the unsaturated fatty acids in the sphingomyelin (SM) fraction is NA at both time points, so NA can be considered as being the key building component in myelin membrane sphingolipids [4].

In their pioneer study, Babin et al. suggested that erythrocyte NA levels in the SM fraction might be used as an indicator of brain maturity [5]. They reported that, regardless of the type of infant feeding, NA values in erythrocyte SM lipids of preterm (PT) infants (32 weeks of gestation) increased steadily until the theoretical term age (37 weeks), suggesting thereby its effective metabolization from oleic acid (C18:1n-9, OA) and incorporation into membrane SM lipids [5]. Similarly, full-term (FT), healthy infants also exhibited significantly increased serum phospholipid NA levels between 2nd days and 4th months of age, meanwhile NA levels in human milk (HM) significantly decreased [6].

Investigations on fatty acid (FA) concentrations in the developing brain revealed that while the major LCPUFAs, AA, and DHA showed a rapid increase in the phosphatidylethanolamine fraction, NA also increased very rapidly in the SM lipids during the first eight years of life, an observation further supporting the important role of NA in myelination [7].

In a recent animal study, LCMUFA metabolite-rich (total C20:1n-9 + C22:1n-9 + C24:1n-9 content about 35%) plant-based oil (*Acer truncatum* Bunge seed oil) supplementation to six-week-old Sprague-Dawley rats resulted in improved cognitive function and brain remodeling [8]. These experimental data support the role of NA and/or its precursors in the structural development of the brain and in the maturation of brain functions such as learning and memory.

The potential developmental role of not only n-3 and n-6 LCPUFAs but that of n-9 MUFAs was also addressed in HM compositional studies. When HM samples were collected daily from the 3rd to the 30th days postpartum in eight healthy Chinese mothers, not only NA but the precursor n-9 metabolites, C20:1n-9, and erucic acid (EA, C22:1n-9) values were found to be highest in colostrum (C), with significantly decreasing contribution over ongoing lactation [9].

We were able to find only six such studies comparing HM from mothers of PT and FT newborns during the first month of lactation, where n-9 FA metabolites longer than OA were also reported [10,11,12,13,14,15]; however, we were unable to find any day-to-day approach of HM sample collection. This paucity of information prompted us to re-analyze our available data on changes in the FA composition of LCMUFAs (C20:1n-9, EA, and NA) in early HM. Our aim was to investigate the changes in these FAs in both PT and FT HM samples during lactation and to determine whether there are differences between PT and FT HM.

## 2. Materials and Methods

### 2.1. Analytical Method

In the databases of our previous studies on HM [16,17,18,19], we re-analyzed previously individually unpublished LCMUFA (C20:1n-9, EA, and NA) data. The detailed process of sample collection and processing is described in the respective articles, but the most important steps are also described here. In the study of Molnár et al. [16] C was collected at the 5th day, while mature HM (MHM) at around 4th month (1–14 months) of lactation with a 24 h collection and pooling of the HM of each mother. After each breastfeeding, the mothers set aside 4 mL of milk and stored it in the refrigerator, and the next day, after 24 h of collection and pooling, it was transported to our laboratory. The samples were stored in a −20 °C freezer until processing. In the study of Minda et al. [17] manually expressed hind milk was collected in the morning (between 08:00–10:00) each day during the first week, then on the 14th and 28th day of lactation and stored in a refrigerator (4–8 °C) for less than four hours, when it was transported to our laboratory. The samples were stored in a −20 °C freezer until processing. In the study of Kovács et al. [18] manually expressed hind milk samples were collected from the evening breastfeeding each day during the first week of lactation, then on the 14th, 21st and 28th days of lactation and stored in a refrigerator (4–8 °C) until transport to our laboratory early next morning. The HM samples were stored in a −80 °C freezer until processing. In the study of Mihályi et al. [19] 5 mL C was collected on the first day of lactation, then 5–10 mL HM sample at the 6th week and 6th months of lactation. Milk samples were stored in a refrigerator (4–8 °C) until transport to our laboratory next day. The HM samples were stored in a −80 °C freezer until processing. All samples were thawed only once, immediately before analysis. In all studies, lipid was extracted from 100 μL milk sample with chloroform/methanol, and internal standard (pentadecanoic acid, C15:0) was also added. Fatty acids were hydrolyzed in HCl and trans-esterified with methanol. In all our HM studies, fatty acid methyl esters were determined by a Finnigan 9001 gas chromatograph (Finnigan/Tremetrics Inc., Austin, TX, USA) by a flame ionization detector and we used either 40 m [16,17,18] or 60 m [19] long cyanopropyl columns (DB-23; J&W Scientific, Folsom, CA, USA). The detailed temperature program and other settings were published in the original publications. FA values are expressed as the weight% of total FAs (data are median (Q3–Q1)).

### 2.2. Statistical Analysis

Statistical analysis of FA data was performed by IBM SPSS Statistics 28.0 software (IBM Corporation, Armonk, NY, USA). Mann-Whitney U test was used to assess the difference between groups and Wilcoxon Signed Rank Test to check differences within groups. To provide a better overview of the changes in FA composition, the data of these three LCMUFAs were mathematically aggregated to a so-called estimated “summarized LCMUFA value”.

Logarithm (Ln) trendline was fitted with both PT and FT HM data of all LCMUFA metabolites and summarized LCMUFA values as well.

## 3. Results

All studies [16,17,18,19] included only healthy lactating mothers of apparently healthy, singleton newborns. Even in the study on HM after PT delivery, PT infants also appeared to be healthy [18]. In all studies, the maternal age was around 30 years and the average maternal BMI was in the normal range (Table 1). In the first study, maternal weight and BMI were significantly higher in the C group compared to the MHM group [16]. In the PT HM study PT babies had significantly lower gestational age, birth weight and birth length compared to the FT group, but maternal characteristics didn’t differ significantly [18]. Except for the PT group, newborn babies had normal birth weight, birth length and gestational age. In the included studies, more mothers were multipara than primipara.

In one of our previous studies [16], we published FA composition of term HM samples taken at two time points of lactation: C (5th day of lactation) and mature HM (MHM, mean sampling time: 4th month of lactation). All LCMUFA (C20:1n-9, EA, and NA) values were higher in the C than in MHM (median [IQR] 0.39 [0.24] vs. 0.35 [0.12], 0.11 [0.09] vs. 0.05 [0.02] and 0.15 [0.10] vs. 0.05 [0.02], respectively). In another of our studies on Hungarian breastfeeding mothers [19], we compared FA composition in HM at three time points: 1st day of lactation (C), 6th week, and 6th month of lactation (both MHM). All LCMUFAs (C20:1n-9, EA, and NA) showed significantly higher values in C than in MHM [19].

### 3.1. Day-to-Day Changes in LCMUFAs in FT HM during the First Week of Life

In our previous communication on changes in the FA composition of HM during the first month of lactation [17], we focused on changes in saturated and polyunsaturated FAs and on the correlations between various polyunsaturated fatty acids, but we did not report all data on MUFA isomers.

Not only for each of the three individual LCMUFA isomers (Figure 1a) but also for the calculated summarized LCMUFA values (Figure 1b) we found a significant decrease with increasing duration of lactation. For each of the monounsaturated fatty acids (C20:1n-9, EA, and NA), the highest values were found in C, followed by decreasing values in transitional milk (TM) and MHM.

### 3.2. Comparison of Day-to-Day Changes in LCMUFAs in PT and FT HM during the First Week of Life

In another of our previous studies, we compared FA compositional changes in HM samples from mothers delivering PT or FT infants [18]; however, only C20:1n-9 was reported from the group of LCMUFA metabolites.

Values of the most important LCMUFA, NA were about three-fold higher in PT than in FT HM samples at all the 10 comparatively investigated time points of lactation (Figure 2c). Moreover, we found similar changes in the shorter chain precursors of NA, i.e., in C20:1n-9 (Figure 2a) and in EA (Figure 2b). The highest NA values were measured on the first days of lactation in both PT and FT groups. By the 21st day of lactation, the contribution of NA to PT HM decreased to a level similar to that seen in the FT HM on the first day of lactation. Ln trendline fitted to both FT and PT data of LCMUFA metabolites and summarized total LCMUFA values (Figure 3) showed highly significant inverse associations between the availability of these FAs and the duration of lactation.

## 4. Discussion

We present hitherto unpublished LCMUFA data for HM samples of Hungarian mothers participating in four previous studies. In all these studies, C20:1n-9, EA, NA, and summarized LCMUFA values were the highest in C, on the first day of lactation, and significantly decreased during the course of lactation. Moreover, LCMUFA values were two- to three-times higher in PT than in FT milk.

LCMUFA metabolites do not have a generally accepted classification in the literature, some authors classify FA isomers with carbon numbers 20 and 22 as LCMUFA [20,21,22], while others classify FA isomers with carbon numbers between 20, 22, and 24 as “Σ20:1n-9, 22:1n-9, 24:1n-9” or VLCMUFA [23]. In this study, we used LCMUFA to cover all the longer-chain metabolites of C18:1n-9, namely C20:1n-9, EA, and NA.

The incorporation of the most important LCPUFAs, AA, and DHA into neuronal tissues occurs mostly in the last trimester of pregnancy and the initial postnatal months. The exclusive dietary source for these FAs is HM for breastfed babies; previous studies showed that HM lipid content, lipid composition, and FA composition of the various lipid classes change over lactation [24,25]. Moreover, significant differences were found in these parameters among the mothers of very preterm (VPT), PT, and FT neonates [26,27]. As to LCMUFA, a recent study found a positive correlation between gestational age at birth and C20:1n-9 and EA content in the HM in VPT babies, while the lactation stage had significant negative correlations with C20:1n-9, EA, and NA values [28].

Pooled data analysis from 55 English-language articles [29] reported the FA profile of HM along lactation stages after PT and FT delivery. Seven articles compared the FA composition of HM in both PT and FT samples; however, only three of these articles [12,13,14] included two or more LCMUFAs (C20:1n-9, EA, and NA). Despite the diverse methods of the included studies (e.g., different gestational ages, times of sampling, number of participants), all three LCMUFAs decreased over time in HM of mothers of both PT and FT neonates (Appendix A). We also calculated the estimated summarized LCMUFA values for each group at each time point addressed in this article [29]. In contrast to our findings, none of the individual or summarized LCMUFA values were higher in PT than in FT milk (Appendix A). Nevertheless, it should be noted that these data are only of limited comparability with our results, because different data from different articles at different time points (for C, TM, and MHM) and in PT and FT groups with different gestational ages were pooled.

By reviewing the literature, we were unable to find any day-to-day follow-up study examining the FA composition of PT and FT HM. Most studies sample within a shorter time period but different authors use different definitions of lactation stages (C, TM, MHM), so when summarizing the results, these different time periods may overlap in time. Some studies (Table 2) have a relatively wide sampling period, and the distribution of lactation periods is not uniform, with considerable overlap between publications, e.g., TM (4–7 day) sampling period by Bobinski et al. [11] meets with that of C (≤7 day) by Thakkar et al. [15]. Moreover, some [10,11,13,14,15] but not all [12] studies followed the same mothers during lactation.

Reviewing the published literature to date, we found a total of four studies comparing HM samples from mothers who delivered PT and FT newborns at all three time points (C, TM, MHM), and all three LCMUFA values (C20:1n-9, EA and NA) were included in the results (Table 2). Aydin et al. [10] did not find statistically significant differences in long-chain n-9 metabolites neither between FT and PT milk samples nor with the progression of lactation. In a Spanish study relatively short sampling ranges with well-separated periods [13] were applied, and despite the small group numbers (VPT, *n* = 10; PT, *n* = 10 and FT, *n* = 23), a statistically significant decrease in the individual values of LCMUFAs (C20:1n-9, EA and NA) from C to MM for all three groups were found. Compared to our results, no statistically significant differences were found between the PT and FT groups in the investigated time points.

Another Spanish research group published the earliest study we could find [14], and looked at the differences in fatty acid composition between HM samples from mothers who gave birth to six PT newborns and mothers who gave birth to 16 FT newborns. Possibly due to the very small number of PT groups, no differences in individual fatty acid values were found either with advancing lactation or between PT and FT groups. Although there were no significant differences among the lactation stages, in both groups the highest C20:1n-9, EA, and NA values were measured in the C samples (1–5 days). In our studies with a day-to-day approach, we saw a significant decline in the individual LCPUFA values with advancing lactation time, so a big standard deviation may arise from this large sampling period. In our previous studies, both EA, and NA in MHM could be detected (between 0.01–0.06 *w*/*w*%), but this Spanish research group [14] failed to determine these fatty acids in the MHM samples of the PT group because it decreased below the limit of quantification.

In a more recent study [15] the fatty acid composition of HM from PT (*n* = 27) and FT (*n* = 34) mothers in Switzerland was analyzed over time. The collection periods of C and TM samples were well separated, but the time interval of sampling of MHM (2–16 weeks of lactation) was extremely wide. In contrast to our results, they found significantly higher values in the FT group compared to the PT group in EA and NA values, while similar to our results the C20:1n-9 values were significantly higher in the TM in the PT group. Similar to our results all published single LCMUFAs (20:1n-9, EA, NA) in this study decreased over lactation in both PT and FT groups, although they didn’t analyze statistically all the fatty acid changes over lactation.

In our present study, we found significantly higher C20:1n-9, EA, NA, and summarized LCMUFA values in PT than in FT HM samples at all the time points investigated. However, the literature on differences in LCMUFA values between PT and FT HM is far from being unequivocal. Higher values of C20:1n-9 in TM [15] or in MHM [30] than in PT were reported. Moltó-Puigmartí et al. [13] found significantly higher LCMUFA values in the VPT TM samples but significantly lower in the C samples compared to FT HM samples, whereas no differences between PT and FT groups were seen. On the other hand, a number of studies reported significantly lower C20:1n-9 values in C [12,30] or MHM [11,31] in PT samples compared to the FT group. In a Swiss study [15], EA and NA values were also significantly lower in the PT than in FT samples. In other studies, no differences in C20:1n-9, EA, and NA values between the PT and FT groups were reported for C [10,14], TM [10,14,15], and MHM [10,13,14,15] samples.

We also observed a significant decline in C20:1n-9, EA, and NA and summarized LCMUFAs values in the course of lactation in both PT and FT groups. This reduction is more clearly in concert with the results of previous surveys. In the studies investigating VPT [13,28] or PT milk samples, a significant decrease in C20:1n-9 [12,13], EA [12,13], and NA values [13,14,32] during lactation were reported. Similar decreases in C20:1n-9 [9,12,13,15,25,33,34,35,36], EA [9,12,13,15,33,34,36], and NA values [9,13,15,25,33,34,36] were also seen in FT samples with the progression of lactation. Only a few reports suggested no change in C20:1n-9 [37,38], EA [25,39], and NA [24,38,39] values with ongoing breastfeeding. Both mixed donor HM [40] and HM substitute infant formula [9,35] have significantly lower C20:1n-9 and NA content compared to PT or FT C and TM samples, respectively.

In preterm infants, lower NA values were found in plasma phospholipids at the first week of age in small for gestational age than in appropriate for gestational age neonates, and gestational ages significantly correlated with C24:1n-9 plasma levels. Moreover, healthy preterm babies at the first week of age had significantly higher NA plasma levels, and NA concentration at one month corrected age was correlated positively to some psychomotor and mental development indices [41]. However, the role of NA and the other LCMUFAs in neurodevelopment is much less investigated than the role of n-3 and n-6 fatty acids, further studies are clearly needed to clarify the possible importance of these FAs in the perinatal period.

Not only gestational age or stage of lactation, but geographical location of sampling can also influence values of the availability of C20:1n-9 and EA [42,43]: the lowest values were reported in the Philippines and highest in China [43]. Significant differences were reported also among different regions in the same country [42], suggesting that the availability of LCMUFAs might be influenced by maternal diets. For instance, the traditional Chinese diet of the Chongqing province is rich in eggs, chicken, and pork resulting in considerably higher C20:1n-9 and EA values compared to other Chinese (e.g., Hong Kong) or international (e.g., Canada) HM samples [42]. Interestingly, in HM samples of women living in Chongqing, LCMUFA values did not decrease with the advancing duration of breastfeeding, but increased significantly, reaching their highest values by the 8th week of lactation [42].

As to the source of NA in humans, in their pioneer animal study Fulco et al. [44] suggested that although lignoceric acid (C24:0) is completely synthesized from acetate, NA is not derived from C24:0 by a simple desaturation step, but rather from OA after repeated chain elongation steps. Another animal study showed that maternal supplementation with canola oil containing NA resulted in an increase in NA content in milk samples, as well as in both heart and liver tissue samples of pups after two weeks of suckling [45]. However, it is still unclear if the main source of NA in the human infant is HM or whether endogenous synthesis from OA may contribute to adequate LCMUFA supply in the early postnatal period of life.

Our study has several strengths. In each study, except for Molnár et al. [16], we followed the same mothers during the lactation, so the effect of interindividual diversity on the FA status among mothers could be minimalized. With the day-to-day approach, we were able to track changes for each LCMUFA more sensitively than with the rarer sampling frequency seen in several previous studies. A further strength is that we determined all n-9 long-chain metabolites (C20:1n-9, EA, and NA) and were able to calculate summarized LCMUFA values (sum of C20-24 MUFAs) for all samples and time points.

Our study has also several limitations. The number of included mothers was usually low (8 to 18 mothers), except in the study of Mihályi et al. (*n* = 87) [19]. The studies included in this re-evaluation were performed over a longer time period (from 2002 to 2015), so there might be dietary or lifestyle differences among the participants. Furthermore, the values of analytical determinations relatively far apart in time can only be compared with some caution, although the trend of changes was very similar across all studies.

## 5. Conclusions

Although the role of NA in the perinatal period is much less clear than those of AA and DHA, our results suggest the possible importance of n-9 LCMUFAs in the early postnatal period. In this study we found significantly higher C20:1n-9, EA, NA, and summarized LCMUFA values in PT than in FT HM samples at almost all investigated time points, and C20:1n-9, EA, NA, and summarized LCMUFA values decreased significantly in the course of lactation. Further studies are needed to define the role of NA and the other LCMUFAs in the diet of preterm and full-term infants.

## Figures and Tables

**Figure 1 life-13-01205-f001:**
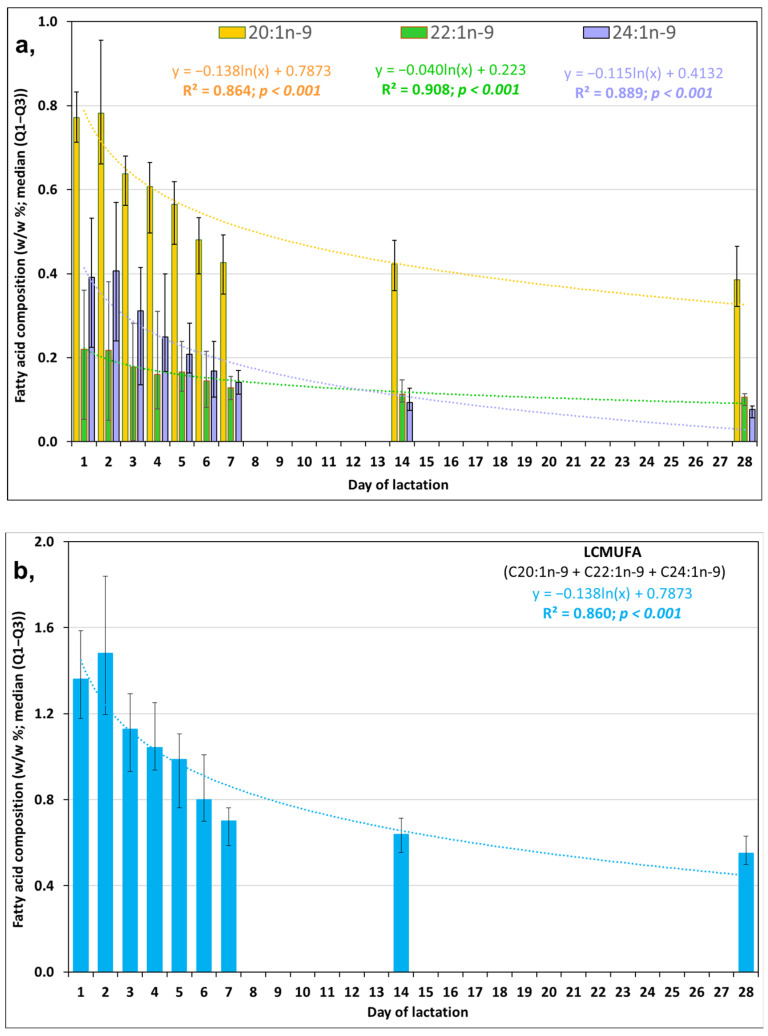
(**a**) Contribution of gondoic acid (C20:1n-9), erucic acid (C22:1n-9), and nervonic acid (C24:1n-9) as well as (**b**) summarized total long-chain monounsaturated fatty acid (LCMUFA) values in human milk samples obtained from mothers of full-term infants (*n* = 18) [17].

**Figure 2 life-13-01205-f002:**
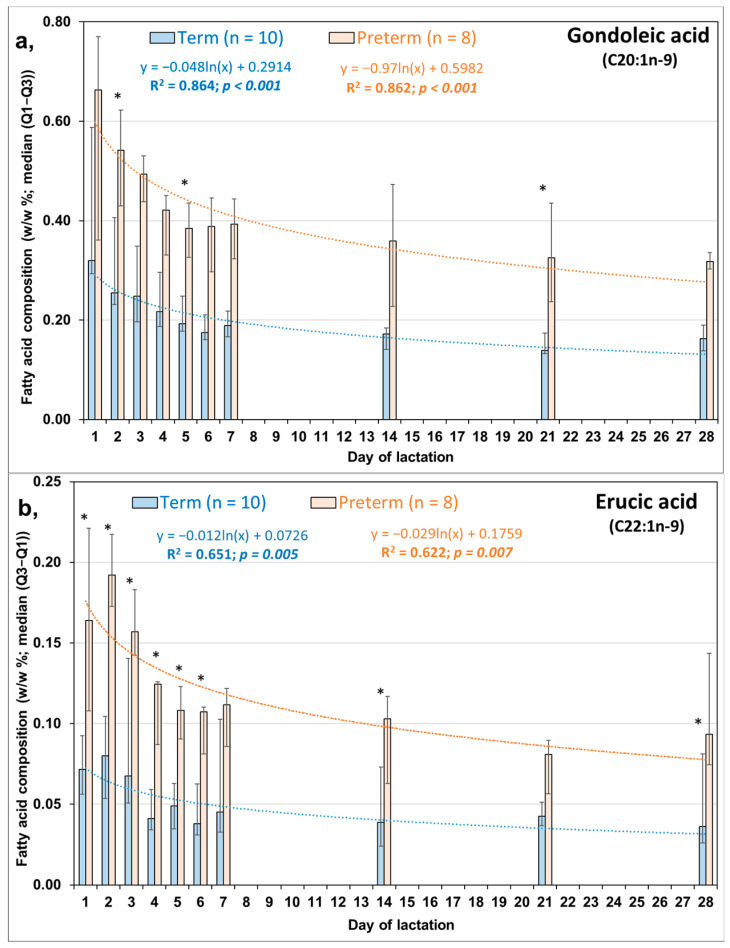
Contribution of (**a**) gondoic acid (C20:1n-9), (**b**) erucic acid (C22:1n-9), and (**c**) nervonic acid (C24:1n-9) to the fatty acid composition of human milk obtained from mothers of full-term (*n* = 10) and preterm (*n* = 8) infants [18] (*: asterisk denotes significant differences between term and preterm milk, Mann-Whitney U test, *p* < 0.05).

**Figure 3 life-13-01205-f003:**
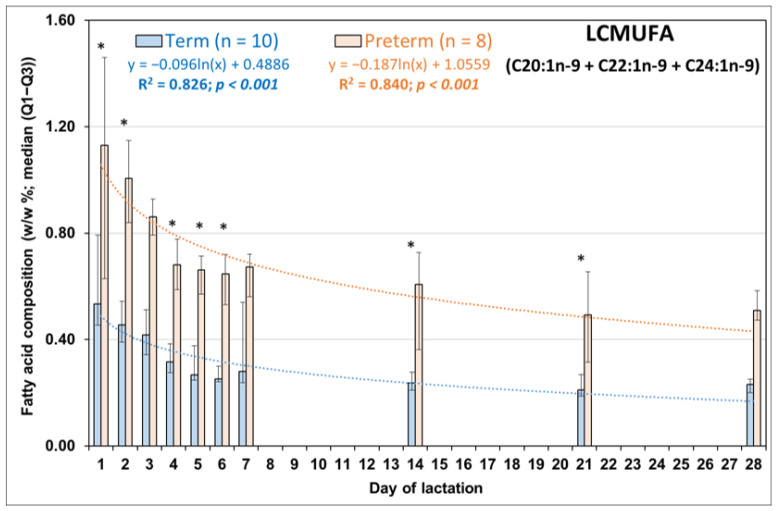
Contribution of summarized total long-chain monounsaturated fatty acids (LCMUFA) to the fatty acid composition of human milk obtained from mothers of full-term (*n* = 10) and preterm (*n* = 8) infants (*: asterisk denotes significant differences between full-term and preterm milk, Mann-Whitney U test, *p* < 0.05).

**Table 1 life-13-01205-t001:** Study characteristics of the included mothers and their newborns.

	Molnár S, 2002 [16]		Minda H, 2004 [17]	Kovács A, 2005 [18]		Mihályi K, 2015 [19]
Number of included mothers	*n* = 18	*n* = 15	*n* = 18	*n* = 10	*n* = 8	*n* = 46
Maternal age (years)	n.d.	n.d.	29.4 ± 4.0	28.0 (4.5)	30.5 (4.2)	32.9
Maternal weight (kg)	68.4 ± 12.0 ^a^	60.2 ± 6.2 ^a^	60.5 (23.6)	64.75 (8.7)	60.0 (7.0)	n.d.
Maternal BMI (kg/m^2^)	24.7 ± 3.2 ^a^	22.2 ± 3.3 ^a^	22.2 (6.8)	24.3 (5.4)	22.0 (3.5)	n.d.
Gestational age (weeks)	*	*	39.1 ± 1.6	38.5 (2.7) ^b^	28.0 (4.2) ^b^	>37th week
Birth weight (g)	*	*	3537 ± 528	3375 (282) ^b^	1235 (420) ^b^	3535 ± 517
Birth length (cm)	*	*	51.3 ± 2.8	50.5 (2.5) ^b^	36.0 (4.7) ^b^	50.7 ± 2.3
Parity (primipara/multipara)	n.d.	n.d.	6/12	2/8	1/7	21/26

Data are either mean ± standard deviation or median (interquartile range). *: denotes normal (uncomplicated), full term pregnancy with healthy newborns; n.d.: denotes: no published data in the original publication. ^a^: *p* < 0.05; ^b^: *p* < 0.001.

**Table 2 life-13-01205-t002:** Individual LCMUFA (C20:1n-9, C22:1n-9, C24:1n-9) values in human milk samples at different lactation stages after preterm (PT) and full-term (FT) delivery in former studies.

Weight%		C20:1n-9		C22:1n-9		C24:1n-9	
Study	Lactation Stage	PT	FT	PT	FT	PT	FT
Thakkar et al., 2018 [15]	C (≤7 day)	0.76	0.76	0.19	0.25	0.30	0.39
	TM (8–14 day)	0.60	0.54	0.19	0.12	0.13	0.13
	MM (15–112 day)	0.47	0.45	0.09	0.08	0.07	0.07
Rueda et al., 1998 [14] *	C (1–5 day)	0.92	0.80	0.20	0.32	0.37	0.44
	TM (6–15 day)	0.69	0.69	0.18	0.18	0.27	0.18
	MM (16–35 day)	0.57	0.54	n.d.	0.09	n.d.	0.08
Moltó-Puigmartí et al., 2011 [13] *	C (2–4 day)	0.96	0.96	0.28	0.27	0.30	0.32
	TM (8–12 day)	0.58	0.52	0.14	0.12	0.11	0.08
	MM (28–32 day)	0.48	0.47	0.10	0.10	0.05	0.05
Aydin et al., 2014 [10]	C (3rd day)	1.39	1.21	0.34	0.33	0.72	0.78
	TM (7th day)	1.23	1.05	0.3	0.25	0.77	0.74
	MM (28th day)	1	0.86	0.2	0.17	0.64	0.37

Fatty acid values are given in the median form, except for *, which is in the mean form. C: colostrum, TM: transitional milk, MHM: mature human milk, FT: full term, PT: preterm. For the chronological classification of delivery and milk sampling, we have followed the original articles, so there may be an overlap between groups.

## Data Availability

The data presented in this study are available upon request to the corresponding author.

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
