# Peer review of "Higher Availability of Long-Chain Monounsaturated Fatty Acids in Preterm than in Full-Term Human Milk"

_life, 2023, doi:10.3390/life13051205_

Round 1
Reviewer 1 Report
Paper is well written.
Study characteristics table is missing.
Author Response
Thank you for your kind review and suggestion, we included the study characteristics table (Table 1 in the Results section).
Reviewer 2 Report
In the manuscript Higher Availability of Long-chain Monounsaturated Fatty Acids in Preterm than in Full-term Human Milk, the authors investigated the changes in monounsaturated fatty acids (gondoic acid (C20:1n-9), erucic acid (EA), and nervonic acid (NA)) composition in preterm (PT) and full-term (FT) human milk samples during lactation.
The article is well written and informative and lays the ground for future studies aiming to define the role of monounsaturated fatty acids in the diet of preterm and full-term infants.
Author Response
Thank you very much for your time, effort and kind words on the review.
Reviewer 3 Report
The study includes an important issue. But there are some shortcomings;
method section: Milking method should be specified; expressing by hand or pump; The time of milking, the way of milking (foremilk-hind milk) and amount should be written; storage conditions should be given.
results section: *maternal age, number of pregnancies, duration of pregnancy, mode of delivery, chronic disease and diabetes in the mother should be given.
*Table S2 is absent.
Author Response
Thank you for your work and kind suggestions. According to the original publications we summarised sample collection methods and storing conditions for each included study in the Methods section.
In the Results section we also included study characteristics of the included studies (Table 1) and in the main text we also described the investigated mothers and their neonates. All studies included only healthy lactating mothers (no chronic disease) of apparently healthy newborns after a normal pregnancy. Even the preterm babies appeared to be healthy, only born prematurely.
Unfortunately, Table S2 was accidentally omitted from the Supplementary files, but taking into account the suggestions of the other reviewer, it was finally included in the main text (Discussion part) instead of its original place.
Reviewer 4 Report
Dear Authors,
The manuscript “Higher Availability of Long-chain Monounsaturated Fatty Acids in Preterm than in Full-term Human Milk” is generally very well written and contains data of some relevance for a general readers as well as of high relevance for specialists in the topic. Although the subject of the paper could be of interest for the readers of the journal, the paper needs some corrections.
Strengths of the paper:
Fairly extensive discussion of the results compared to the rest of the manuscript.
Weaknesses of the paper:
Generally, I think that the manuscript should be treated as a review article. Little research has been done. Some of the results of these studies have been partially presented in other publications. In my opinion, this article would be more interesting if more literature was gathered and if the results obtained in previous works on long-chain monounsaturated fatty acids were compared with those of other authors. Alternatively, the manuscript can be treated as “communication”.
In my opinion, the method should be described in more detail - for example, the method of fat extraction, the exact parameters of the chromatographic analysis.
Author Response
Thank you for your kind review and suggestions. We added to the Methods section the milk collection methods for each previous studies and the lipid extraction method also. According to your suggestion we completed the Discussion section with a point-by-point comparison, where we compared our results with the results of all the articles published so far where all three LCMUFA (C20:1n9, C22:1n-9 and C24:1n-9) values that we examined from PT and FT breast milk samples were analyzed in one article.
We believe that our article is more than a review, as we are examining the role of nervonic acid and other LCMUFAs in breast milk during the lactation based on our four previously published articles. We also publish here new results, as in the article of Kovács et al only C20:1n-9 was indicated and in Minda et al no LCMUFA values were described. Although in Molnár et al all three LCMUFA values can be found, this article is in Hungarian and it is questionable whether it can be found and downloaded from abroad. Moreover, using the original data we also calculated the summarized LCMUFA values and made new statistical evaluation of the results. I hope you will find these arguments acceptable. Thank you again for your great suggestions.
Round 2
Reviewer 1 Report
Authors have extensively revised the manuscripts suggested.
Reviewer 3 Report
Accept in the present form
Reviewer 4 Report
Dear Authors,
Thank you for answering my questions and completing the manuscript. If the Editors have no objections, I can accept the manuscript as well. The work is well developed in terms of content and in its current form the discussion of the results is extensive. So I can accept the paper in present form.